# The Design and Test of the Chassis of a Triangular Crawler-Type Ratooning Rice Harvester

**Weijian Liu** [1,2], **Xiwen Luo** [1,2], **Shan Zeng** [1,2,*], **Li Zeng** [1,2] and **Zhiqiang Wen** [1,2]

[1] Key Laboratory of the Ministry of Education of China for Key Technologies for Agricultural Machine and Equipment, South China Agricultural University, Guangzhou 510642, China; weijianliu@stu.scau.edu.cn (W.L.); xwluo@scau.edu.cn (X.L.); lizeng@stu.scau.edu.cn (L.Z.); cm25032@stu.scau.edu.cn (Z.W.)

[2] College of Engineering, South China Agricultural University, Guangzhou 510642, China

[*] Correspondence: shanzeng@scau.edu.cn; Tel.: +86-20-3867-6975

**Abstract:** Due to the high rolling rate of a regular crawler paddy harvester and the absence of mature first season harvester products of ratooning rice, combined with the planting mode and harvest requirements of ratooning rice, a triangular crawler ratooning rice harvester is specifically designed. The structure and steering principle of the triangular crawler chassis are described. The hydraulic system is simulated and analyzed by AMESim2020 (Guangzhou, China) to verify the rationality of its design; RecurDynV9R4 (Guangzhou, China) is used to simulate and analyze the field straight/turning situation of differential steering chassis and rear-axle steering chassis. The results show that the rear axle steering chassis has a smaller turning radius and lower rolling loss rate and the change of track tension is more stable during steering. The field test is conducted to verify the reliability of the simulation results. The field test shows that the rolling loss rate of the rear axle steering chassis is reduced by 27.9% compared with the differential steering chassis. The machine's operating speed is 2.8 km/h, the minimum turning radius is 780 mm, and the straight rolling rate is 26.8%. The operating performance is stable, and the operational process is smooth. Compared with the existing conventional harvester, the linear rolling rate of the first harvest of ratooning rice is reduced by 26.1%, and the test results are consistent with the RecurDyn simulation results. The results are reliable, providing a reference for the theoretical research of the chassis of the later ratoon rice harvester.

**Keywords:** ratooning rice; triangular crawler; turning radius; track tension; virtual prototype; text





## 1. Introduction

Rice is an important staple food crop globally, whose harvesting is an essential issue in its production line [1]. Due to the advantages of low cost and high benefit, the ratooning rice plantation area is increasing year by year [2–6]. In the first season, the mechanized harvest of ratooning rice needs to ensure high stubble retention and low rolling rate [7–9]. If the ordinary crawler harvester is used, the rolling rate is more than 50%, seriously affecting the yield of ratooning rice.

Rice combine harvesters in developed countries such as Europe and the United States have been popularized and applied on a large scale, but there has been no detailed research on ratooning rice harvesters, and ratooning rice harvesters in China are still in the primary stage. Zhang Guozhong of Huazhong Agricultural University and others have developed a kind of ratooning rice-cutting machine. The machine adopts a seedling transplanter chassis and narrow paddy field wheel, which has a low rolling rate for ratooning rice, but significant subsidence, challenging field driving and steering, and a low degree of mechanical automation [10]. Therefore, the researchers have developed a dual-channel feeding ratooning rice harvester, mainly composed of a crawler chassis, header, two sets of threshing and symmetrically arranged cleaning devices left and right, straw crusher, grain

tank, and power and transmission system. The cutting width is 3 m and the field mobility is poor [11]. In this case, the complete simulated analysis results of crawler ratooning and the rolling rate of the walking chassis of the paddy harvester conducted asfield testing is reported in [12]. However, the research and development of ratooning rice harvesters are still in the primary stage.

Due to the above problems, we used the multi-body dynamics software Recur-DynV9R4 (Guangzhou, China) to simulate and analyze the ratooning rice harvester's field turning motion from the perspective of kinematics and dynamics. The constraints and motion relationship between various components deduce the force relationship between the track and the ground, the track, and each gear train, determine the optimal chassis structure through theoretical calculation and simulation analysis, and conduct field tests to verify the reliability of the simulation results. It provides a basis and reference for the subsequent ratooning rice harvester development.

## 2. Overall Design of the Ratooning Rice Harvester

### 2.1. Design Requirement

Field water content is high during the first harvest of ratooning rice, and the paddy field operation environment is complex. It requires high chassis power and strong trafficability. At the same time, it also needs to meet the requirements of low rolling rate, small crawler settlement, high ground clearance, and low turning radius. The triangular crawler chassis can further improve the ground clearance of the chassis under the conditions of ensuring small grounding specific pressure, good mobility, and flexible steering. Therefore, it is widely used in hilly, mountainous, and paddy fields [13–15].

This paper puts forward the design scheme of a triangular crawler chassis: (1) the triangular crawler walking mechanism is adopted, the paddy field trafficability is good, the crawler with width of 260 mm is adopted, and the straight rolling rate is low; (2) a full hydraulic drive is adopted, with a simple structure and a reliable performance, and a 0~4.5 km/h stepless speed regulation is adopted to meet the stable variable speed operation under various working conditions and realize the rapid conversion between machine field operation and road transportation; (3) four wheel drive, good trafficability, and anti-soil overturning performance; and (4) small turning radius.

### 2.2. Overall Structure and Working Principle

The triangular crawler chassis of a ratooning rice harvester comprises a frame, walking system, power and transmission system, and relevant accessories [16]. Working parts, such as the threshing and cleaning device, conveying trough, header, grain tank, and grain unloading drum, are installed above the frame, as shown in Figure 1.

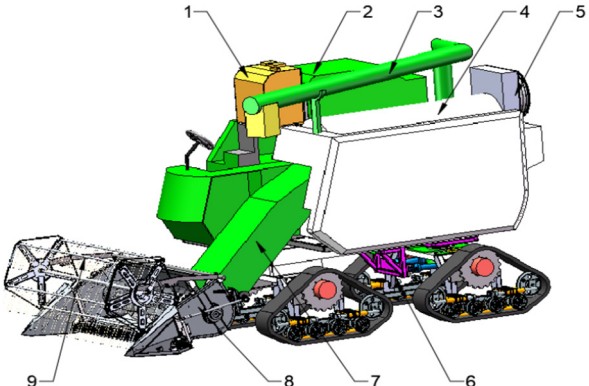

**Figure 1.** Triangle caterpillar reproducing rice harvester. (1) Hydraulic oil tank (2) Grain tank (3) Grain unloading drum (4) Threshing and cleaning device (5) Radiator (6) Chassis (7) Conveying trough (8) Header (9) Reel.

Hydrostatic, mechanical, and hydro-mechanical transmission are the most common transmission modes for rice harvesters [17–19]. The operation of the mechanical transmission is complex, and for harvesters with a wide range of power variations, more gears are required to meet the requirements of the corresponding operations [20]. Furthermore, standard gear transmission cannot realize step-less speed change [21]. Therefore, the entire hydraulic pressure drives the ratooning rice harvester's triangular crawler chassis. The main pump converts the mechanical energy output by the engine into hydraulic energy. The hydraulic energy is transmitted to the hydraulic traveling motor to drive the triangular crawler wheel to complete the forward and backward actions. Table 1 shows the main technical parameters of the triangular crawler chassis of the ratooning rice harvester.

**Table 1.** Main technical parameters of triangular crawler chassis for ratooning rice harvester.

| Parameter | Value |
|---|---|
| Overall dimension (L × W × H)/(mm × mm × mm) | 5300 × 2680 × 3150 |
| Mass/(kg) | 4500 |
| Power/(kW) | 74.5 |
| Driving mode | Four wheel drive |
| Track width/mm | 1500 |
| Wheelbase/mm | 1800 |
| Track width/mm | 260 |
| Track grounding length/mm | 800 |
| Minimum ground clearance/mm | 600 |

### 2.3. Kinematic Analysis of a Triangular Crawler Chassis

As shown in Figure 2, the equation of motion is:

$$\triangle x_m = x_m^{k+1} - x_m^k = l_c cos\left(\theta_m^k + \alpha^k\right) \tag{1}$$

$$\triangle y_m = y_m^{k+1} - y_m^k = l_c sin\left(\theta_m^k + \alpha^k\right) \tag{2}$$

$$\triangle \theta_m = \theta_m^{k+1} - \theta_m^k \tag{3}$$

where $x_m^k/x_m^{k+1}$ is the coordinate of the centroid $x$ direction of the crawler chassis at the time of $k$ and $k + 1$, m; $y_m^k/y_m^{k+1}$ is the coordinate of the track chassis centroid $x$ direction at the time of $k$ and $k + 1$, m; $\theta_m^k/\theta_m^{k+1}$ is the direction angle of the crawler chassis at time $k$ and $k + 1$, rad; $\alpha^k$ is the included angle between the track forward direction and motion collection, rad; $\triangle x_m$ is the displacement of the chassis centroid in the $x$ direction, m; $\triangle y_m$ is the displacement in the $y$ direction of chassis centroid, m; and $l_c$ is the distance between $o_m^k$ and $o_m^{k+1}$, m.

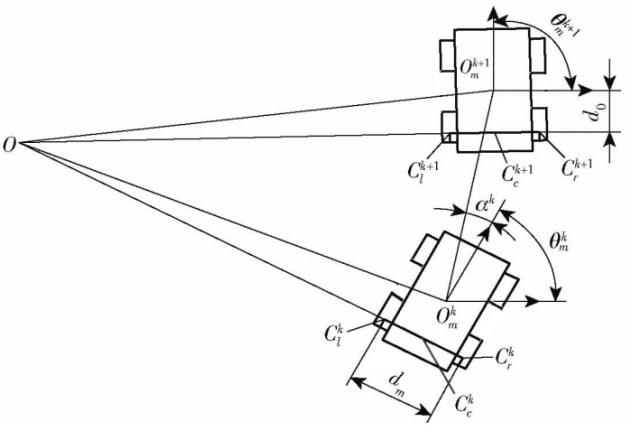

**Figure 2.** Motion analysis of triangular crawler chassis.

The chassis coordinates are defined as:

$$Z = (x_m, y_m, \theta_m) \tag{4}$$

The derivation of Equation (4) gives:

$$\dot{Z} = \begin{bmatrix} \frac{r\cos\theta_m}{2} - \frac{d_0 r \sin\theta_m}{d_m} & \frac{r\cos\theta_m}{2} + \frac{d_0 r \sin\theta_m}{d_m} & \frac{r\cos\theta_m}{2} - \frac{d_0 r \sin\theta_m}{d_m} & \frac{r\cos\theta_m}{2} + \frac{d_0 r \sin\theta_m}{d_m} \\ \frac{r\sin\theta_m}{2} + \frac{d_0 r \cos\theta_m}{d_m} & \frac{r\sin\theta_m}{2} - \frac{d_0 r \cos\theta_m}{d_m} & \frac{\sin\theta_m}{2} + \frac{d_0 \cos\theta_m}{d_m} & \frac{\sin\theta_m}{2} - \frac{d_0 \cos\theta_m}{d_m} \\ \frac{r}{d_m} & -\frac{r}{d_m} & \frac{1}{d_m} & -\frac{1}{d_m} \end{bmatrix} \begin{bmatrix} q_r \\ q_l \\ s_{rx} \\ s_{lx} \end{bmatrix} \tag{5}$$

where $r$ is the radius of track driving wheel, mm; $d_m$ is track gauge, mm; $d_0$ is the vertical distance from the mass center of track chassis to the centerline of track driving wheels on both sides, mm; $q_r$ is the angle of the right track driving wheel, rad; $q_l$ is the angle of the left crawler drive wheel, mm; $s_{rx}$ is the sliding displacement of the right track, mm; and $s_{lx}$ is the sliding displacement of the left track, mm. Equation (5) can calculate the moving speed and angular speed of the triangular crawler chassis.

### 2.4. Theoretical Analysis of Steering Performance

During differential steering, it is possible to feel the turn with a different radius by changing the speed of the track driving wheels on both sides [22–24]. The lateral bulldozing resistance of the crawler increases the friction resistance between the surface and the soil increases. With the increase of the settlement of the crawler, the ability of the track spikes to damage the soil intensifies, and the phenomenon of lateral bulldozing becomes more apparent. The lateral movement of the crawler destroys the rice root system, resulting in a significant increase in the rolling rate. Therefore, this paper plans to use a hydraulic power-assisted rear axle steering chassis to reduce the lateral movement of the rear crawler. During steering, the rear axle rotates around the shaft under the push of the rear axle oil cylinder. When the track wheel on one side has no speed, and the trackwheel on the other side has speed, it is unilateral braking steering. At this time, the lateral bulldozing capacity of the rear track is weakened, which can effectively reduce the rolling rate. Figure 3 shows the illustration of steering.

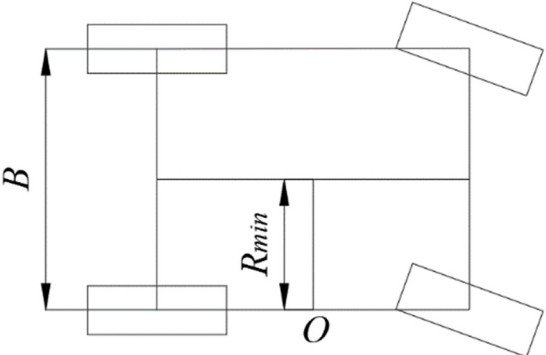

**Figure 3.** Schematic diagram of triangle caterpillar chassis steering of ratooning rice harvester.

According to the geometric analysis, the minimum turning radius is:

$$R_{min} = \frac{B}{2} \tag{6}$$

where $R_{min}$ is the minimum turning radius, mm; $B$ is the triangular crawler chassis track of ratooning rice harvester, mm. The triangular crawler-type chassis wheelbase ratooning rice harvester is 1500 mm. Replace Equation(8) to obtain the minimum turning radius of 750 mm.

### 2.5. Tension Calculation of Triangular Crawler

During the movement of the triangular crawler chassis, the driving wheel outputs the driving torque, and the friction will be generated between the bearing wheel, guide wheel, and the contact surface of the crawler to hinder the movement of the crawler. Under the influence of the tensioning device, the crawler will be affected by the dynamic tensioning force. The track wheel divides the track into multiple sections. The calculation formula of dynamic tension at the front and rear of a track wheel is as follows:

$$F_q = \begin{cases} k_1(b_{k-1} - b_0) & (b_{k-1} > b_0) \\ F_0 & (b_{k-1} = b_0) \end{cases} \tag{7}$$

$$F_w = \begin{cases} k_1(b_{k-1} - b_0) & (b_{k+1} > b_0) \\ F_0 & (b_{k-1} = b_0) \end{cases} \tag{8}$$

where $F_q$ is the tension force on the front side of the track wheel, N; $F_w$ is the tension force on the rear side of the track wheel, N; $k_1$ is the tension force on the rear side of the track wheel, N; $b_{k-1}$ is the tension force on the rear side of the track wheel, N; $b_{k+1}$ is the tension force on the rear side of the track wheel, N; $b_0$ is the tension force on the rear side of the track wheel, N; $F_0$ is the tension force on the rear side of the track wheel, N.

Considering the interaction between the track and the ground, the dynamic tension on the front and rear sides of a track wheel are:

$$F_w - F_q = T_w t \tag{9}$$

where $T_w$ is the tangential force component of the track wheel, N; $t$ is the unit vector. It can be seen from the above analysis that the dynamic tension force of the crawler is affected by the tensile direction stiffness and deformation length of the crawler. The former is determined by the structure and material of the crawler, whereas the latter is affected by the force of the tensioning device, field road conditions, the speed of the ratooning rice harvester, and many other factors.

## 3. Hydraulic System Design

### 3.1. Hydraulic Schematic Diagram

The hydraulic system ensures that the rotational speeds of the hydraulic traveling motors on both sides are same when driving under different road conditions, to avoid the deviation of the ratooning rice harvester. Speed adjustment can be realized under different working conditions to meet the requirements of walking speed. Figure 4 illustrates a hydraulic drive system used to support the above functions.

Volume speed regulation has no throttling loss and overflow loss, and the system has high efficiency. It is suitable for high-power systems and widely used in heavy machinery, agricultural machinery, and other fields [25,26]. Therefore, the hydraulic system adopts the open system of variable pump motor mode. The system uses a single variable pump to operate the four variable motors, with the belt-drive connected to the power output terminal of the leading pump engine and the hydraulic travelling motor connected to the triangular crawler driving wheel. A coupling connects the pilot pump with the primary pump. The main pump outputs working pressure oil, and the pilot pump outputs pilot pressure oil. Operate the foot valve to connect the pilot pressure oil and push the spool of the multi-way directional valve to slide, which can change the flow direction of the working pressure oil and flow into the hydraulic walking motor from different oil inlets to realize forward rotation or reverse rotation and drive the machine forward and backwards. The two hydraulic traveling motors on the left and right sides are connected in parallel. The working pressure oil flows into the two hydraulic traveling motors on the same side after 1:1 shunting through the gear shunting motor, ensuring the synchronization of the hydraulic traveling motors on the same side. Part of the pilot pressure oil output by the pilot

pump flows into the full hydraulic steering gear. When the driver turns the steering wheel, it drives the steering gear valve core to rotate, and the hydraulic oil enters the steering cylinder to realize steering. When the steering wheel turns, it drives the steering gear valve core to rotate, and the hydraulic oil enters the steering cylinder to realize steering.

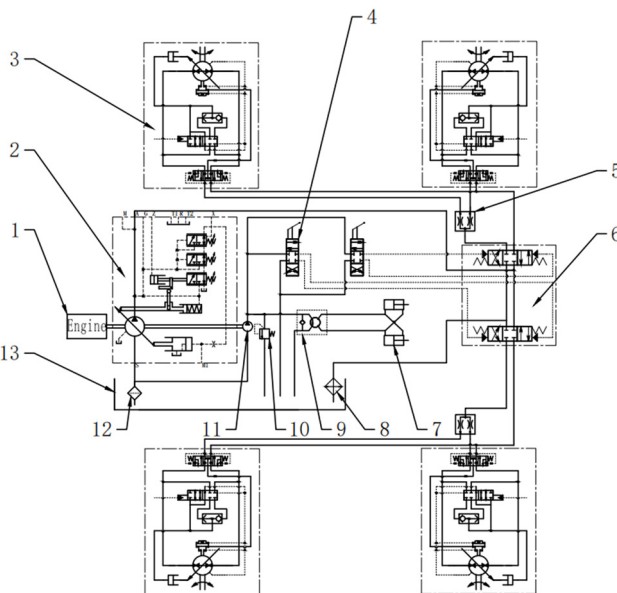

**Figure 4.** Hydraulic system schematic diagram. (1) Engine (2) Main pump (3) Hydraulic traveling motor (4) Foot valve (5) Gear shunt motor (6) Multi way directional valve (7) Steering cylinder (8) Radiator (9) Steering gear (10) Overflow valve (11) Pilot pump (12) Filter (13) Oil tank.

*3.2. Calculation and Selection of Main Hydraulic Components*

3.2.1. Calculation of Hydraulic Traveling Motor

The ratooning rice harvester has the maximum resistance under the climbing condition of full load. At this time, the maximum resistance is:

$$F_z = (m + m_z)f_1 g + (m_z + m)gf_2 + (m + m_z)gsin\alpha \tag{10}$$

where $F_z$ is total load climbing resistance, N; $m$ is the mass of the whole machine, kg; $m_z$ is the load mass, kg; $\alpha$ is the maximum climbing angle, 30°; $f_1$ is the rolling resistance coefficient, 0.12; and $f_2$ is the friction resistance coefficient, 0.1.

The torque of a hydraulic traveling motor is:

$$M_m = \frac{F_z R}{n\eta_1 \eta_2} \tag{11}$$

where $M_m$ is the torque of a single hydraulic traveling motor, N·m; $R$ is the radius of the driving wheel, m; $n$ is the number of hydraulic traveling motors; $\eta_1$ is the mechanical efficiency of hydraulic travel motor, 0.9; and $\eta_2$ is the efficiency of triangular crawler wheel, 0.9.

The theoretical displacement of hydraulic traveling motor is:

$$V_m = \frac{2\pi M_m}{P_m \eta_3} \tag{12}$$

where $V_m$ is the displacement of the hydraulic traveling motor, mL/R; $P_m$ is the system's working pressure, 20 MPa; and $\eta_3$ is the volumetric efficiency of the hydraulic travel motor, 0.94. The hydraulic traveling motor shall meet the requirements of the maximum traveling

speed of the triangular crawler type ratooning rice harvester, so the maximum speed of the hydraulic traveling motor is:

$$n_{max} = \frac{V_{max}}{2\pi R} \tag{13}$$

where $n_{max}$ is the maximum speed of hydraulic traveling motor, r/min and $V_{max}$ is the maximum traveling speed of the harvester, m/s. By substituting the design parameters into Equations (10)–(13), the hydraulic travelling motor adopts LTM04A resolution by consulting the corresponding hydraulic motor models with a theoretical range of = 838 mL/r and maximum speed = 50 r/min. Table 2 shows its key parameters.

**Table 2.** Main parameters of hydraulic walking motor.

| Displacement (mL/r) | Theoretical Output Speed (r/min) | Maximum Working Pressure (MPa) | Maximum Output Torque (N·m) |
|---|---|---|---|
| 738/1265 | 54.18/31.6 | 21 | 4200 |

### 3.2.2. Calculation of Hydraulic Pump

The output flow and theoretical displacement of the hydraulic pump are:

$$Q_p = \frac{4k_p V_m n_{max}}{1000} \tag{14}$$

$$V_p = \frac{1000 Q_p}{n\eta_1\eta_2} \tag{15}$$

where $Q_p$ is the output flow of the hydraulic pump, L/min; $k_p$ is the leakage coefficient, 1.1; $V_p$ is the theoretical displacement of the hydraulic pump, mL/r; $n_f$ is the speed of hydraulic pump, R/min; and $\eta_4$ is the volumetric efficiency of the hydraulic pump, 0.94. By substituting the design parameters into Equations (14)–(15), the output flow of the hydraulic pump is 184.4 L/min, and the theoretical displacement is 78 mL/r. Table 3 illustrates the main characteristics of the A11VO75LRDS axial piston variable displacement pump, selected as the hydraulic pump, using the samples of the relevant hydraulic pump.

**Table 3.** Main parameters of hydraulic pump.

| Displacement (mL/r) | Nominal Pressure (MPa) | Peak Pressure (MPa) | Limit Speed (r/min) |
|---|---|---|---|
| 85 | 35 | 40 | 2500 |

### 3.2.3. Engine Calculation

The driving power of hydraulic pump is:

$$P_e = \frac{\gamma Q_p P_p}{60\eta_5\eta_6} \tag{16}$$

where $P_e$ is the engine power, kW; $p_p$ is the pressure of hydraulic pump, MPa; $\gamma$ is the conversion coefficient, 0.4; $\eta_5$ is the mechanical efficiency of the hydraulic pump, 0.9; and $\eta_6$ is the belt transmission efficiency, 0.9. According to Equation (16), the engine power = 53 kW. The Changchai 4G33TC engine with a rated power of 74.5 kW and a rated speed of 2600 r/min is selected when considering the power reserve.

### 3.3. Hydraulic System Simulation

The system simulation model was developed in AMESim2020(Guangzhou, China) to simulate the control accuracy of the system under different working conditions including straight-line driving, braking, and turning [27–29].

### 3.3.1. Simulation of Hydraulic System when Driving in a Straight Line

Set the engine speed as 2000 r/min, the motor displacement as 37 mL/r, the displacement of the variable displacement pump gradually increases from 0, and the simulation time is 10 s. Calculate the torque acting on one side of the crawler when the ratooning rice harvester travels in a straight line in the field, and bring the parameters into the load to obtain the simulation diagram of the straight line driving condition of the crawler ratooning rice harvester in the field shown in Figure 5.

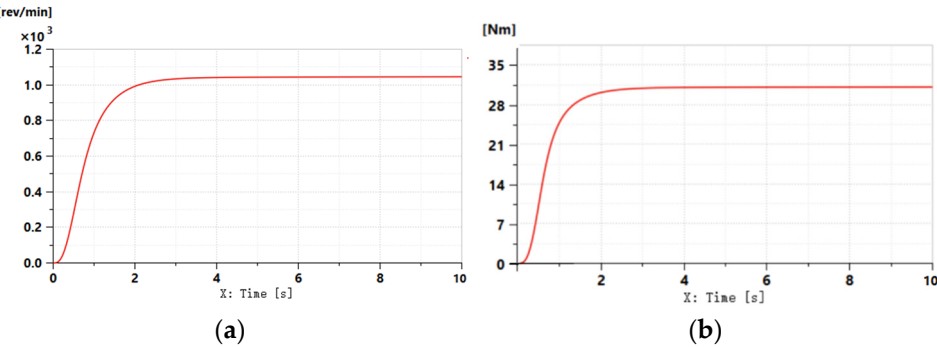

(**a**)                                                                 (**b**)

**Figure 5.** Hydraulic system schematic diagram: (**a**) hydraulic motor speed and (**b**) hydraulic motor torque.

It can be seen from Figure 5a that the speed of the hydraulic motor is stable at about 1040 r/min in 2.5 s. According to the calculation, the traveling speed of the crawler type ratooning rice harvester is about 3 km/h. It can be seen from Figure 5b that the speed output and torque output of the hydraulic motor reach stability after 2.5 s. Multiplied by the reduction ratio, the output torque of the hydraulic walking motor is 692.3 N·m. The simulation results are close to the calculation results of RecurDynV9R4 (Guangzhou, China).

### 3.3.2. Simulation of Hydraulic System during Braking of the Ratooning Rice Harvester

After the motor speed of the ratooning rice harvester is stable, the speed of the harvester is slowed down. During the deceleration process, the maximum displacement of the variable pump is reduced to 0 slowly, and the braking time is 1 s. The harvester decelerates under the joint action of the motor braking torque and soil driving resistance. The output torque, output speed, and output port flow of the hydraulic horse are simulated and analyzed. Figure 6 shows the braking simulation results.

Figure 6 shows that the displacement of the variable displacement pump decreases to 0 within 1 s, the speed of the hydraulic motor decreases to 0, and the speed of motor decreases more stable one. The output braking torque of the motor is approximately 61 N·m, which corresponds to the actual parameters of the variable displacement pump. The simulation model is reasonable and can accurately reflect the working conditions of various pump-controlled motor hydraulic system components.

### 3.3.3. Simulation of Hydraulic System in Field Turning

In order to simulate the turning condition of the ratooning rice harvester in the field, fully open the foot valve on one side and half-open the foot valve on the other side. Figure 7 shows the simulation results.

Figure 7 shows that the opening size of the foot valve is positively related to the motor's output speed. When the engine speed does not change, the speed of the hydraulic motor can adjust by changing the opening size of the foot valve to realize the field difference steering of the ratooning rice harvester.

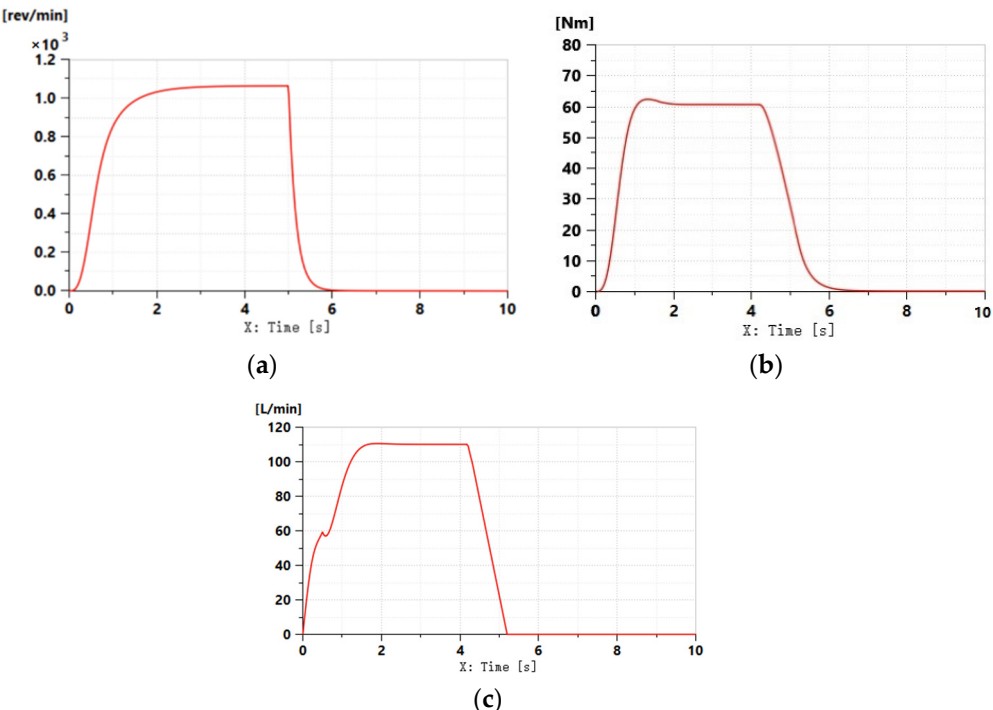

**Figure 6.** Braking simulation analysis: (**a**) hydraulic motor speed, (**b**) hydraulic motor torque, and (**c**) variable displacement pump outlet flow.

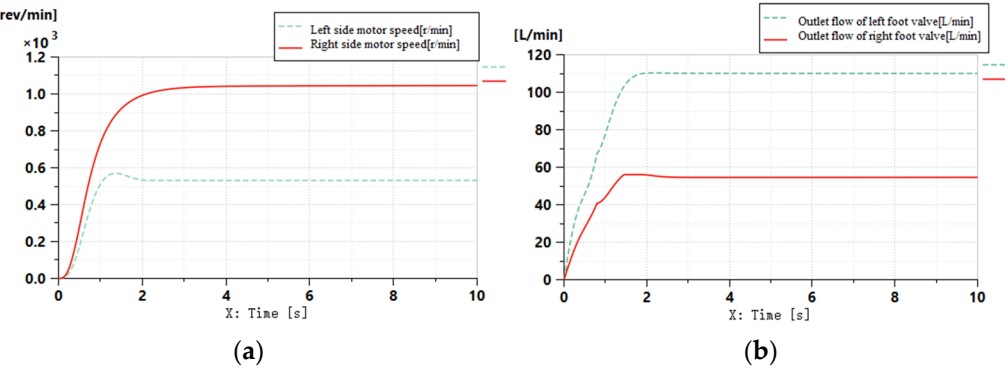

**Figure 7.** Simulation analysis of field-turning conditions: (**a**) hydraulic motor speed, (**b**) variable displacement pump outlet flow.

## 4. Model Establishment

CATIAV5R20 (Guangzhou, China) and RecurDynV9R4 (Guangzhou, China) were used to design the virtual prototype of the ratooning rice harvester. In the monitoring subsystem, the driving wheel, support wheel, and track support are fixed and controlled to determine the key and motion relationships of the various components [30–32]. Each track subsystem is equipped with a driving wheel, four groups of load wheels, and two tensioning wheels. The driving wheel has a radius of 194.5 mm, a landing length of 800 mm, a track gauge of 1600 mm, and a track width of 260 mm for the construction of the monitoring subsystem. Except for the track system, the rest of the virtual prototype does not participate in the operating system. The rests are imported directly from CATIAV5R20 (Guangzhou, China) to RecurDynV9R4 (Guangzhou, China) as a solid body structure. All-wheel trains support and tracks form a tracking subsystem, and track subsystems on either side can set the road contact parameters [33–35]. The turning radius of the ratooning rice harvester adds movement to the driving wheel in rotating pairs and defines and controls the driving wheel's speed by the step (time, $T_0$, $Y_0$, $T_1$, $Y_1$) function. Furthermore, it can

operate the wheels on both sides, attach the auxiliary pulley to the mechanical body with the rotating pair, move the relative position of the tensioning wheel, and adjust the tension of the track, respectively.

The differential steering chassis and rear axle steering chassis are simulated and analyzed. Group I adopts a differential steering chassis, and group II adopts rear axle steering chassis. The maximum steering angle of rear axle is $20°$. The two groups of simulation tests set the same driving wheel motion parameters. Figure 8 shows the virtual prototype model.

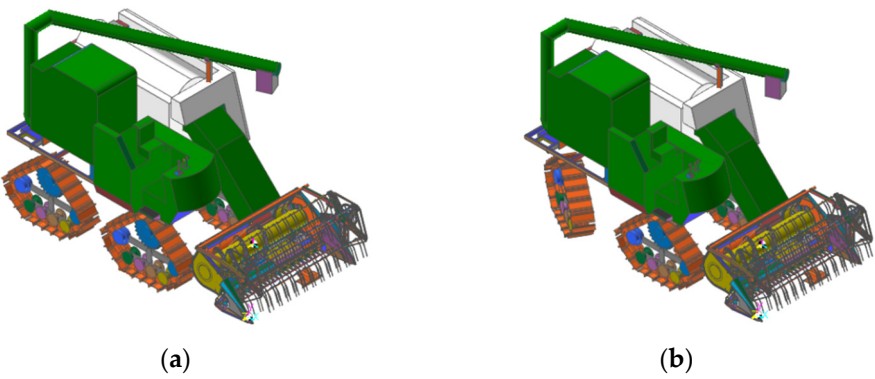

(**a**)  (**b**)

**Figure 8.** Virtual prototype model: (**a**) differential steering chassis and (**b**) rear axle steering chassis.

*4.1. Establishment of the Pavement Spectrum Model*

The standard random process theory established the simulation pavement model. The vertical profile of the road and crosswalk with the road surface consideration is the model of roughness, which describes by two numbers of power spectral density of the model. The principle of the harmonic superposition method is to assume that the pavement elevation is a stable and ergodic Gaussian process, which different forms of trigonometric series can simulate. We used the harmonic superposition method to establish the simulation pavement model.

The power spectral density of pavement displacement can be expressed as:

$$G(n) = G(n_0)\left(\frac{n}{n_0}\right)^{-2} \tag{17}$$

$G(n)$ is the power spectral density of pavement displacement; $G(n_0)$ is the coefficient of road roughness under the reference spatial frequency $n_0$; and $n$ is the spatial frequency, $m^{-1}$. Using the characteristics of stationary random process, the road roughness variance between spatial frequencies $n_1$ and $n_2$ is:

$$S^2 = \int_{n_1}^{n_2} G(n)dn \tag{18}$$

Further, the frequency space between $n_1$ and $n_2$ is divided into $k$ cells, the center frequency of each cell is $n_m$, and the corresponding road spectrum unevenness between the K cells is expressed as: using the characteristics of stationary random process, the road roughness variance between spatial frequencies $n_1$ and $n_2$ is:

$$q(x) = \sqrt{2G(n_m)}sin(2\pi n_m x + \theta_k) \tag{19}$$

where $G(n_m)$ is the road roughness coefficient corresponding to the $k$-th cell; $\triangle n_k$ is the length of the kth cell; and $\theta_k$ is the phase angle. It is possible to obtain random displacement input by superimposing the sinusoidal functions between all cells.

## 4.2. Simulation Analysis of Straight Line Driving in Field

According to the field operation environment, build solid pavement and field pavement models in RecurDynV9R4 (Guangzhou, China). Table 4 shows the characteristic parameters of the two kinds of pavement.

**Table 4.** Main parameters of ground.

| Type | Parameter | Value |
|------|-----------|-------|
| Solid pavement | Ground stiffness/$(N·m^{-1})$ | 10,000 |
| | Damping coefficient/$(N·s·m^{-1})$ | 10.004 |
| | Maximum friction coefficient | 0.45 |
| | Pavement deformation index | 2 |
| Field pavement | Cohesion modulus $K_C$/$(kN/m^{n+1})$ | 42.538 |
| | Internal friction modulus $K_\Phi$/$(kN/m^{n+2})$ | 9.004 |
| | Soil deformation index | 0.8227 |
| | Soil moisture content/% | 37.3 |
| | Soil firmness/kPa | 93.5 |

## 5. Analysis of Simulation Results

### 5.1. Simulation Analysis of Straight Line Driving in Field

The straight-line driving of the ratooning rice harvester is simulated, and the driving wheel speed is defined through the step function, so that the theoretical driving speed of the ratooning rice harvester is 1.50 m/s, ignoring the influence of track slip on the simulation results. Data is collected once every 0.20 s, and the setup of the simulation step is every 25 s. Figure 9 shows the field driving speed of the ratooning rice harvester.

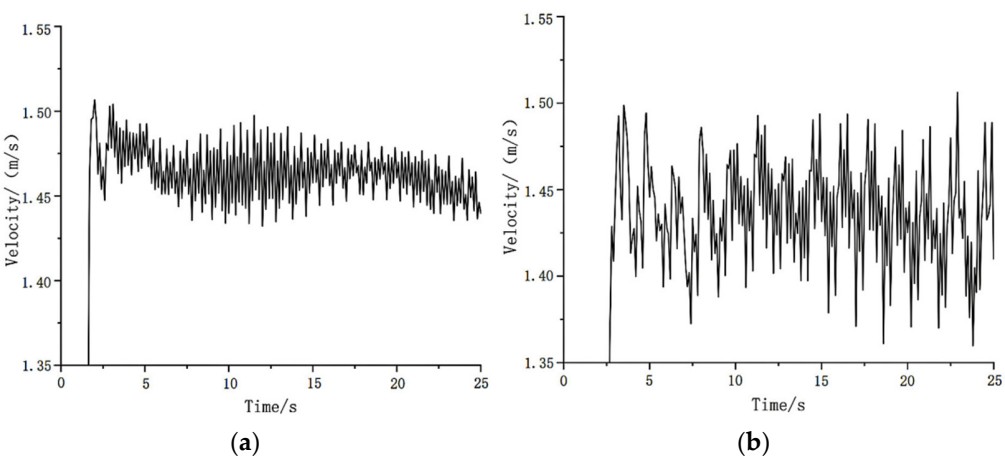

(a)                                    (b)

**Figure 9.** Driving speed simulation results: (**a**) Solid pavement and (**b**) Field pavement.

It can be seen from Figure 9 that when the ratooning rice harvester runs on a concrete road, the speed change is relatively stable, with an average speed of 1.472 m/s, a maximum speed of 1.508 m/s, and a minimum speed of 1.431 m/s; When the ratooning rice harvester runs on the field road, the standard deviation of the speed is significant, and the speed variation fluctuates widely. The average speed is 1.423 m/s, the maximum speed is 1.507 m/s, and the minimum speed is 1.358 m/s. The simulation value of the average driving speed of the ratooning rice harvester is slightly lower than the theoretical calculation value and the simulation phenomenon is in line with the expectation.

### 5.2. Field Turning Simulation Analysis

The sinusoidal curve in Figure 10 is the horizontal displacement curve of the ratooning rice harvester with time. One cycle of the sinusoidal curve represents that the ratooning rice harvester completes a circular motion. The sum of the curve and the sum of the tank

is the turning diameter of the ratooning rice harvester. Thus, it is possible to calculate its turning radius. Set the speed of the right crawler driving wheel to be higher than that of the left wheel, and the ratooning rice harvester turns left. Figure 11 shows that the turning radius of the ratooning rice harvester in group II is significantly smaller than that in group I. Calculate the average value of the above eight groups of turning radius data, and the rear axle steering system can reduce the turning radius of the ratooning rice harvester by 42.3%, which is consistent with the above theoretical analysis. In addition, the simulation results for group II indicate that differential steering has little effect on the turning radius of the ratooning rice harvester, with the turning radius primarily determined by the steering angle of the rear axle.

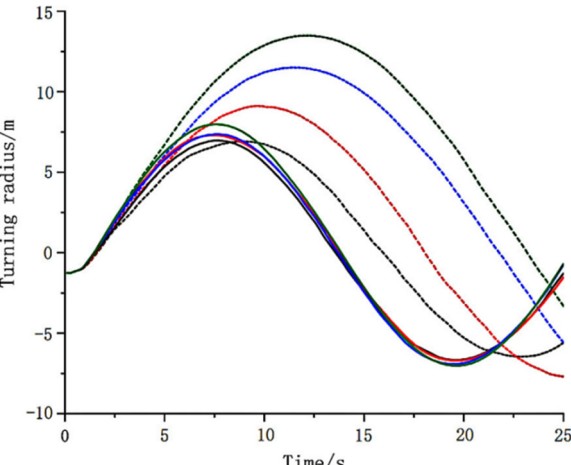

**Figure 10.** Horizontal displacement curve of the ratooning rice harvester. Note: the dotted line refers to group I chassis and the solid line refers to group II chassis; the same color curve represents the same track motion parameters.

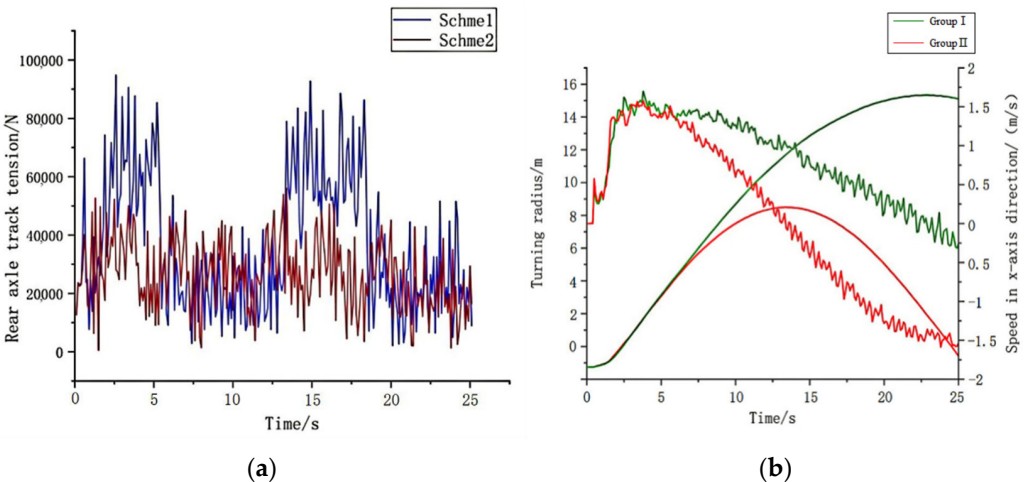

(**a**)                    (**b**)

**Figure 11.** Track tension:(**a**) rear axle track tension and (**b**) turning radius and speed in *x*-axis direction.

Figure 11 illustrates the track tension of the ratooning rice harvester under the current soil simulation environment and the turning radius under the chassis structure.

During steady-state turning, the speed of the inner crawler decreases and the ratooning rice harvester deflects. Under the condition that the motion parameters of the crawler driving wheels on both sides are the same, the turning radius and the crawler tension of group II in Figure 11a are less than that of group I. When the rear axle of the Group I ratooning rice harvester does not deflect, the side of the crawler constantly pushes the soil, and the friction resistance between the crawler surface and the soil increases crawler tension

increases sharply and fluctuates. The sharp increase of the track tension is straightforward to derail the track, resulting in the fracture of the rubber track and the bending of the guide axle. At the same time, by increasing the amount of bulldozing, the capacity of the dangerous soil path is intensified, the amount of track settlement increases, thus damaging a large area of the paddy root system, and the rolling width is approximately 800 mm above the ground length of the rice root system. The maximum deflection angle of the rear axle of group II recycled rice harvester relative to the machine body is up to 20°. When turning, the movement direction of the rear axle is the same as the tangent direction of the turning arc. There is no bulldozing phenomenon on the side of the track. The track tension is slight and the fluctuation is small. The rolling width of the rice pile is only 260 mm, which is 65% lower than that of group I. Figure 11b shows the variation of speed in the *x* direction with time during steady-state turning of the ratooning rice harvester. In Group II, the speed fluctuation in the *x* direction when turning the ratooning rice harvester is slight because Group II has a steering rear axle with a reasonable structure, low vibration, and high constant speed. The RecurDynV9R4 (Guangzhou, China) simulation results show that the rear axle steering chassis has the advantages of low rolling rate and more flexible turning. It is suitable for the ratooning rice harvester, and the simulation movement phenomenon is consistent with the expectation.

## 6. Field Experiment

### 6.1. Test Conditions

The field experiment of the triangular crawler type ratooning rice harvester was carried out in the science teaching and research base of South China Agricultural University in Zengcheng District, Guangzhou, in April 2022, as shown in Figure 12. The experimental area is 0.6 hm$^2$, the rice variety is Meixiangzhan No. 2, the average plant height is 784 mm, and the water content of straw is 71.29%. The field environmental parameters were tested on the general provisions of agricultural machinery (GB/T5256-2008). The soil moisture content of the test field was 31.4% and the soil firmness was 456.3 kPa.

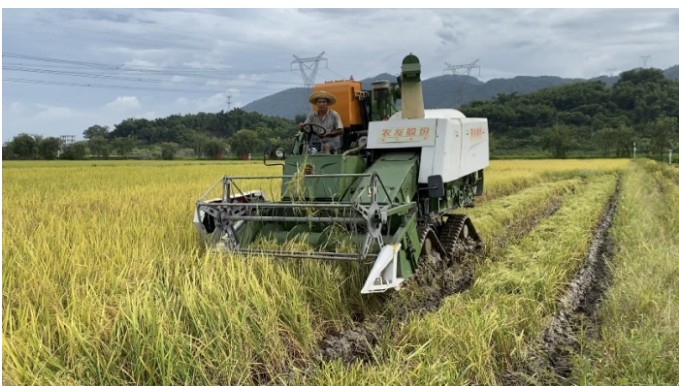

**Figure 12.** Field experiment.

### 6.2. Test Method

The test method for vehicle terrain trafficability (GB/T 12541–1990) and minimum turning radius, minimum turning channel circle diameter and camber value of vehicles (GB/T 12540–2009), harvesting tool turning and rolling loss ratio, driving speed, turning radius and straight such as rolling loss rate, is selected as the primary performance rating indicators.

Turning rolling loss rate: select a 10 m × 10 m area for the test to verify the superiority of the rear axle steering chassis, and the following formula can calculate the rolling loss rate $\eta_s$:

$$\eta_s = \frac{M_{s2}}{M_{s1}} \times 100\% \tag{20}$$

where $\eta_s$ is the rolling loss rate, %; $M_{s1}$ is the track rolling area, m$^2$; and $M_{s2}$ is the passing area of a harvester, m$^2$.

Driving speed: select the paddy field and flat pavement with a length of 30 m, respectively, measure the time required for the prototype to pass through the 30 m measuring area, repeat it three times, and calculate the driving speed.

Turning radius: in the paddy field environment, the prototype runs smoothly at low speed, and the steering wheel remains unchanged when it is at the left/right limit position. The crawler wheel on one side brakes, and the crawler wheel on the other advances. After driving smoothly for 360°, drive out of the test area, and use a steel tape to measure the radius of the rutting track left on the ground in the vertical direction.

Straight rolling loss rate: the Yanmar YH1180 crawler harvester (Yanmar, Wuxi, China) is used as the control group to compare the rolling situation of rice stubble after harvest, and the following formula can calculate the straight rolling loss rate $\eta_L$:

$$\eta_L = \frac{M_{s3}}{M_{s4}} \times 100\% \tag{21}$$

where $\eta_L$ is the rolling loss rate of straight-line driving, %; $M_{s3}$ is the track rolling width, mm; and $M_{s4}$ is the cutting width, mm. and the minimum speed is 1.358 m/s. The simulation value of the average driving speed of the ratooning rice harvester is slightly lower than the theoretical calculation value, but the error is small, and the simulation phenomenon is in line with the expectation.

### 6.3. Test Results and Analysis

Figure 13 compares rolling loss between differential steering chassis and rear axle steering chassis. During the test, the two chassis turn with the same radius. Figure 13a shows the rolling of the differential steering chassis. Four crawlers rotate around the geometric center of the ratooning rice harvester. The lateral bulldozing of the crawler is serious, and the turning indentation area is fan-shaped and large; Figure 13b shows the rolling of the rear axle steering chassis; there is no bulldozing on the side of the track, and the turning indentation area is small. According to the measurement, the turning rolling rate of Figure 13a is 78.3%, the turning rolling rate of Figure 13b (the ratooning rice harvester) is 50.4%, and the turning rolling loss rate is reduced by 27.9%, which is consistent with the simulation results of the virtual prototype.

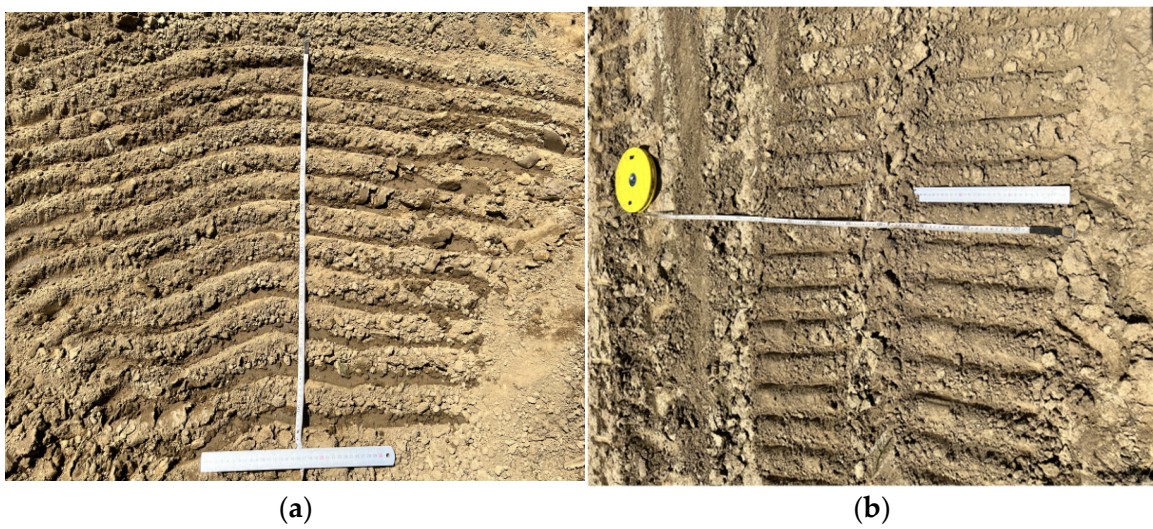

         **(a)**                                      **(b)**

**Figure 13.** Track rolling trace: (**a**) Differential steering chassis and (**b**) Rear axle steering chassis.

Table 5 shows the results of the travelling speed of the ratooning rice harvester test result. The results show that the driving speed of the triangular crawler type ratooning

rice harvester is 0–4.5 km/h, the field operation speed is 0–2.8 km/h, and the minimum turning radius is 780 mm. The field test results can meet the design requirements. The field test results are consistent with the simulation results of the RecurDynV9R4 (Guangzhou, China) virtual prototype. It shows that the simulation results of the virtual prototype are highly reliable and the multi-body dynamics model established has high accuracy.

**Table 5.** Field test results.

| Parameter | Simulation Results | Test Results | Technical Requirement |
|---|---|---|---|
| Driving speed/(km·h$^{-1}$) | 0–4.5 | 0–4.5 | 0–4.5 |
| Operating speed/(km·h$^{-1}$) | 0–2.8 | 0–2.8 | 0–2.8 |
| Minimum turning radius/mm | 748 | 785 | ≤900 |
| Maximum climbing angle/(°) | 233 | 215 | ≥150 |
| Maximum ridge crossing height/mm | 0–4.5 | 0–4.5 | 0–4.5 |

Figure 14 shows the comparison of rolling conditions of the harvester after harvest. After measurement, the straight rolling rate of the control harvester is 52.9%, and the straight rolling rate of the triangular crawler type ratooning rice harvester designed in this paper is only 26.8%, which is reduced by 26.1%.

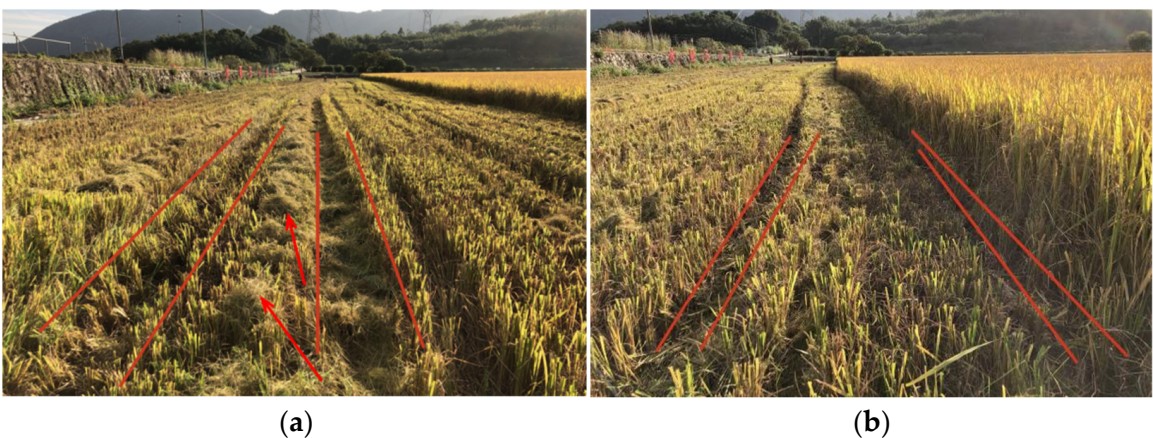

(**a**)          (**b**)

**Figure 14.** Comparison of rolling after harvesting: (**a**) rolling condition of control group and (**b**) rolling condition of ratooning rice harvester.

## 7. Conclusions

(1) The triangular crawler chassis of the ratooning rice harvester is studied and designed in combination with the planting method of ratooning rice in South China and describes and analyzes the overall chassis structure over the working operating principle. The whole machine adopts an entire hydraulic drive, with a minimum ground clearance of 600 mm. The triangular track wheel is equipped with a 260 mm wide track to meet high clearance requirements, low rolling, good trafficability, flexible steering, and simple operation.

(2) The hydraulic drive system is designed. The hydraulic drive system uses a variable pump. The variable motor relies on the gear shunt motor to ensure the synchronization of hydraulic travel motors on both sides and uses a foot valve to flexibly control the hydraulic travel motor's rotation direction and rotation time on either side. Combined with rear-axle steering, the turning radius is small.

(3) The force and motion relationship between various components is analyzed by taking the chassis of a triangular crawler ratooning rice harvester as the research object. A combination of theoretical analysis, modelling simulation, and field testing investigates a triangular crawler's dynamic characteristics. Model simulation and test

verification of differential steering chassis and rear-axle steering chassis are performed based on the harmonic superposition method. The results show that the rear-axle steering chassis has a lower rolling rate and smaller turning radius. The multi-body dynamic model of the chassis of the triangular crawler ratooning rice harvester has high accuracy, which can be used for theoretical simulation analysis and provide a reference for the dynamic analysis and optimization of the crawler chassis.

(4) The field test results show that the rear axle steering chassis can effectively reduce the rolling rate. The operation speed of the machine can reach 2.8 km/h, and the straight rolling rate is only 26.8%. Compared with the traditional crawler rice combine harvester, it has a smaller turning radius and better operation efficiency, and the rolling rate is reduced by 26.1%.

**Author Contributions:** Conceptualization, S.Z. and X.L.; methodology, W.L.; software, Z.W.; validation, W.L., Z.W. and S.Z.; formal analysis, L.Z.; investigation, Z.W.; resources, W.L.; data curation, W.L.; writing—original draft preparation, W.L.; writing—review and editing, Z.W., S.Z. and L.Z.; visualization, Z.W.; supervision, Z.W.; project administration, S.Z.; funding acquisition, S.Z. All authors have read and agreed to the published version of the manuscript.

**Funding:** This research was funded by the Laboratory of Lingnan Modern Agriculture Project (NT2021009).

**Institutional Review Board Statement:** Not applicable.

**Informed Consent Statement:** Not applicable.

**Data Availability Statement:** The data that support the findings of this study are available on request from the corresponding author.

**Acknowledgments:** The authors gratefully acknowledge the editors and anonymous reviewers for their constructive comments on our manuscript.

**Conflicts of Interest:** The authors declare no conflict of interest.

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
