# Peer review of "The Design and Test of the Chassis of a Triangular Crawler-Type Ratooning Rice Harvester"

_agriculture, doi:10.3390/agriculture12060890_

Round 1

Reviewer 1 Report

All comments have inserted in the manuscript.

Author Response

Thank you. I have modified the marks of reviewers in the manuscript

Reviewer 2 Report

Please review the article in writing.

Author Response

Thank you, I have modified the manuscript.

Reviewer 3 Report

This work is about the design of a triangular rice combine harvester. In the work, the Authors described the construction and the driving principle of a triangular tracked chassis. The rationality of the project was confirmed by the simulation method as well as by the field tests. The results were compared with those obtained for a conventional combine harvester.

Author Response

谢谢您的意见。